

# Direct measurements of black carbon fluxes in central Beijing using the eddy-covariance method.

Rutambhara Joshi[1], Dantong Liu [1, *], Eiko Nemitz[2], Ben Langford[2], Neil Mullinger[2], Freya Squires[3], James Lee[3,9], Yunfei Wu[4], Xiaole Pan[4], Pingqing Fu[4], Simone Kotthaus[5,^], Sue Grimmond[5], Qiang Zhang[6], Ruili Wu[6], Oliver Wild[7], Michael Flynn[1], Hugh Coe[1] and James Allan[1,8]

[1]University of Manchester, School of Earth and Environmental Science, Manchester, United Kingdom
[2]UK Centre for Ecology and Hydrology, Penicuik, United Kingdom
[3]Department of Chemistry, University of York, York, United Kingdom
[4]Institute of Atmospheric Science, Chinese Academy of Sciences, Beijing, China
[5]Department of Meteorology, University of Reading, Reading, United Kingdom
[6]Ministry of Education Key Laboratory for Earth System Modelling, Department of Earth System Science, Tsinghua University, Beijing, China
[7]Lancaster Environment Centre, Lancaster University, Lancaster, UK
[8]National Centre for Atmospheric Science, University of Manchester, Manchester, United Kingdom
[9]National Centre for Atmospheric Science, University of York, York, United Kingdom
[*]now at: Department of Atmospheric Sciences, School of Earth Sciences, Zhejiang University, Hangzhou, Zhejiang, China 310027
[^]now at: Institut Pierre Simon Laplace, École Polytechnique, Palaiseau, France

*Correspondence to*: James Allan (james.allan@manchester.ac.uk) and Rutambhara Joshi (joshi.rutambhara@gmail.com)

**Abstract.** Black carbon (BC) forms an important component of particulate matter globally, due to its impact on climate, the environment and human health. Identifying and quantifying its emission sources is critical for effective policymaking and achieving the desired reduction in air pollution. In this study, we present the first direct measurements of urban BC fluxes using eddy-covariance. The measurements were made over Beijing within the UK-China Air Pollution and Human Health (APHH) winter 2016 and summer 2017 campaigns. In both seasons, the mean measured BC mass (winter: 5.49 ng m$^{-2}$ s$^{-1}$, summer: 6.10 ng m$^{-2}$ s$^{-1}$) and number fluxes (winter: 261.25 particles cm$^{-2}$ s$^{-1}$, summer: 334.37 particles cm$^{-2}$ s$^{-1}$) were similar. Traffic was determined to be the dominant source of the BC fluxes measured during both seasons. The total BC emissions within the 2013 Multi-resolution Emission Inventory for China (MEIC) are on average too high compared to measured fluxes by a factor of 58.8 (winter) and 47.2 (summer). Comparison of MEIC transport sector only are also larger (factor winter:37.5, summer: 37.7) than the measured flux. Emission ratios of BC/NO$_x$ and BC/CO are comparable to vehicular emission control standards implemented in January 2017 for gasoline (China 5) and diesel (China V) engines, indicating reduction of BC emissions within central Beijing and extending this to a larger area would further reduce total BC concentrations.



# 1 Introduction

Particular matter with a diameter ≤ 2.5 µm ($PM_{2.5}$) is a major contributor to air pollution (Harrison et al., 1997). It is a global concern given its severe impacts on health, as epidemiological studies identify a variety of cardiovascular and respiratory diseases. (Lelieveld et al., 2015; Pope III and Dockery, 2006). The daily recommended exposure limits of $PM_{2.5}$ suggested by World Health Organisation (WHO) is 25 µg m$^{-3}$ (Janssen and Joint, 2012). However, in China concentrations routinely are order of 100s µg m$^{-3}$ (Li et al., 2017; Zhang et al., 2017; Zíková et al., 2016). Black carbon (BC), in general, contributes up to 10-15 % of overall $PM_{2.5}$ and is emitted from incomplete combustion of fossil fuel and biofuel (Seinfeld and Pandis, 1998). Even though its abundance is relatively low in $PM_{2.5}$, the WHO (2012) report that a 1 µm$^3$ increase in BC is associated with greater health risks compared to the same increase in unidentified $PM_{2.5}$ mass. Furthermore, BC is the most optically absorbent aerosol in the atmosphere, with warming potential to the climate second only to $CO_2$ (Bond et al., 2013; Ramanathan and Carmichael, 2008). Therefore, reducing BC will potentially have major benefits for both air quality and climate.

The capital of China, Beijing, is well known for its air quality issues, with a large population frequently exposed to unsafe levels of air pollution. Therefore, improving air quality is a priority for government and environmental agencies in China. Ongoing policies targeting reduction in the use of solid fuel domestic cooking and heating (Barrington-Leigh et al., 2019.), and introducing traffic management policies and vehicular emission controls (Wu et al., 2017), have been successful in reducing $PM_{2.5}$ and consequently BC in recent years (Sun et al., 2016). However, Beijing still suffers from severe haze episodes, with BC concentrations frequently 10 times greater than observed in western countries (Liu et al., 2014; Wu et al., 2016). Therefore, understanding and quantifying BC emission sources can help reach the Chinese government targets for clean air.

Scientists routinely use atmospheric chemistry models to assess local and regional air quality and determine the effectiveness of potential legislative interventions. As air quality models are underpinned by emission inventories, the accuracy of their forecasts is closely coupled to inventory uncertainties from emission factors and activity statistics (e.g. fuel consumption and source) (Cao et al., 2006; Hodnebrog et al., 2014). Such uncertainties are challenging to resolve as the spatial resolution increases up to few km (Zheng et al., 2017).

In China, sources have rapidly changed with urbanisation and environmental controls. Ground based measurements of emissions can help to refine and verify emission inventories. For BC, this is complicated by the use of BC / $PM_{2.5}$ ratios to estimate BC emissions. This ratio is highly variable depending on fuel type and combustion conditions (Bond et al., 2004; Streets et al., 2001). Consequently, analysis of the physical and chemical properties of BC to help identify source and combustion conditions is essential to better characterise BC particles.

In this study, we measure BC concentrations using a Single Particle Soot Photometer (SP2, DMT, Boulder) and present the first-ever source characterised urban flux measurements of BC determined using the micrometeorological eddy-covariance (EC) method. To date, EC measured BC fluxes have only been undertaken for one grassland site (Emerson et al., 2018). EC allows direct measurement of the magnitude of the net flux from the upwind area, or flux footprint (up to few km,



depending on measurement height and meteorological conditions), providing quantification of the fluxes at the local scale.

This study is part of the wider UK-China Air Pollution and Human Health (APHH) project, which had field campaigns in Beijing during winter (16[th] Nov-10[th] Dec 2016) and summer (15[th] May-30[th] June 2017) (Shi et al., 2019). Also as part of this project, ambient BC concentrations were measured using a separate SP2 connected to a conventional inlet line, as described by Liu et al. (2019) and Yu et al. (2020).

## 2 Method

### 2.1 Site location

During the APHH project, two SP2 instruments were used to study concentrations and fluxes of BC at the Institute of Atmospheric Physics (IAP, 39°58'28"N, 116°22'16"E), sampling from an inlet placed on the IAP meteorological tower. When the tower was built in 1979 for long-term meteorological and environmental monitoring, it was surrounded by croplands. Following rapid urbanisation and industrialisation, it is now surrounded by a heterogeneous urban land use between the 3[rd]

and 4[th] Ring roads. This includes a small park ($\sim 0.3$ km$^2$) to the west, the Beijing-Tibet Expressway ($\sim 400$ m) to the east and a busy road (Beitucheng, crossing from east to west, $\sim$100 m) to the north of the tower.

### 2.2 Instrumentation: SP2

BC concentrations were measured using an SP2 model B (retrofitted with photomultipliers for incandescence detection and the newer 8-channel data acquisition system) during the winter period and an SP2 model C during the summer period. As the same calibration and data analysis protocols were followed, no indication of a systematic difference in quantification was noted. The Droplet Measurement Technologies (DMT) SP2, Boulder, Colorado, can quantify the mass of the refractory BC (rBC) within individual aerosol particles using Laser-Induced Incandescence (LII) without any influence of non-rBC material

(Moteki and Kondo, 2007; Schwarz et al., 2006). The laser beam operating at 1064 nm is used to measure rBC (hereafter referred to as BC) on a single particle basis. The laser heats up the particle causing it to incandesce if BC is present. The intensity of incandescence signal is proportional to the mass of the BC, which requires calibrating using size selected soot standards. In this study, calibrations for both SP2B and SP2C in the incandescence channel were performed throughout each campaign using Aquadag® BC particles, to avoid instruments-based biases of SP2B and SP2C. Since Aquadag® has higher

sensitivity to incandescence channel than ambient BC particles, a correction is applied using a multiplication factor of 0.75 (Laborde et al., 2012). The mass of BC is then converted to mass equivalent diameter using a density of $\rho = 1.8$ g cm$^{-3}$ (Bond and W.Bergstrom, 2006). Here, we use the same terminology as Liu et al. (2014) and refer to the mass equivalent diameter as the core diameter ($D_c$), which is the diameter of the sphere containing the same mass of BC as measured in the particle. The ambient BC is often internally mixed with other aerosol, often referred to as the 'coating'. The overall size of the particle can

be estimated based on the scattering signal using the leading-edge-only (LEO) fitting (Gao et al., 2007). In this study, to



quantify coating content of a BC particle, we introduce a parameter Scattering enhancement ($E_{sca}$) in eq 1, as discussed by Liu et al. (2014) and Taylor et al. (2015).

$$E_{sca} = \frac{S_{measured,coated\ BC}}{S_{calculated,uncoated\ BC}},$$ (1)

Here, $S_{measured,\ coated\ BC}$ is the scattering intensity of the coated BC particle measured using SP2 with LEO fit applied and $S_{calculated,}$
$_{uncoated\ BC}$ is the scattering intensity of the uncoated BC based on the measured mass and using a refractive index of BC of $n_{BC}$ = 2.26–2.16i reported by Moteki and Kondo. (2010) for the SP2 wavelength λ = 1064 nm. $E_{sca}$ is proportional to coating content of a particle and for a given particle if $E_{sca}$ =1, suggests that such particle has no coating, noting that the precision of the scattering measurement and the model used to predict scattering are not perfect, so this measurement should only serve as a qualitative indicator.


## 2.3 Measurement set up

Figure 1 shows the measurement setup for the EC measurements. A 1/2 inch O.D. copper sampling inlet was placed at 102 m height. The measurement height is almost two times higher than the mean building height in all but the southerly direction
(Liu et al., 2012) and it is six times higher than the mean displacement height. The open lattice triangular cross-section tower structure was designed to minimise distortion of the air flow. A three-dimensional sonic anemometer (Model R3-50, Gill Instruments, Lymington, UK) was used during both seasons. The sonic anemometer sampled at a frequency of 20 Hz and it was mounted with a north offset of 13° and 31° during winter and summer measuring periods, respectively. Ambient air was pumped down to the analysers at ground level, with a flow of ~90 L min⁻¹. Additionally, at ground level, the sample line was
isokinetically divided into four flows using a TSI Flow splitter. Two ports were connected to the pump, one was connected to an Aerodyne Aerosol Mass Spectrometer (AMS) which deployed a further bypass pump providing an additional flow of about 1 L min⁻¹ and the other pulled air past the SP2 sampling inlet at a flow rate of ~1 L min⁻¹, from which the SP2 sub-sampled at a flow of 0.1 L min⁻¹. This setup guaranteed turbulent flow profile until just in front of the SP2 and made AMS and SP2 measurements as consistent as possible.

The raw data recorded by the SP2 is the multi-channel data for individual particles, however only a subset of the particles detected are typically recorded, due to limitations imposed by hard drive space and computer throughput. Since we were operating in a heavily polluted environment, we set the particle recording frequency to 1 in 30 for winter and 1 in 5 for summer. Ideally, every single particle (i.e. 1 in 1) would be analysed to maximise counting statistics and minimise uncertainty in the flux calculation but limitations in data storage and processing power of the built-in computer prevented this. Count frequencies
were based on the instrument performance and ambient concentrations, with concentrations higher in winter.

During acquisition, the instrument handles data in 200 ms buffers and each recorded particle is timestamped according to the buffer, so when the data is processed it produces number and mass concentrations with a 5 Hz time resolution. If during acquisition a buffer is not fully processed before the next one has finished collection (which can occur at high concentrations),



then a data buffer is discarded, which is manifested as a 200 ms gap in the data. Overall data losses were 7.5 % in winter and
4.2 % in summer. The higher losses in winter were because of the higher concentrations and the fact the SP2-B used an older
and lower specification logging computer compared to the newer SP-C used in the summer. This is in spite of the lower data
saving frequency. In both seasons, single data gaps were interpolated as they were assumed to be due to dropped buffers.
Longer gaps (Symptomatic of other instrument problems) were treated as downtime.

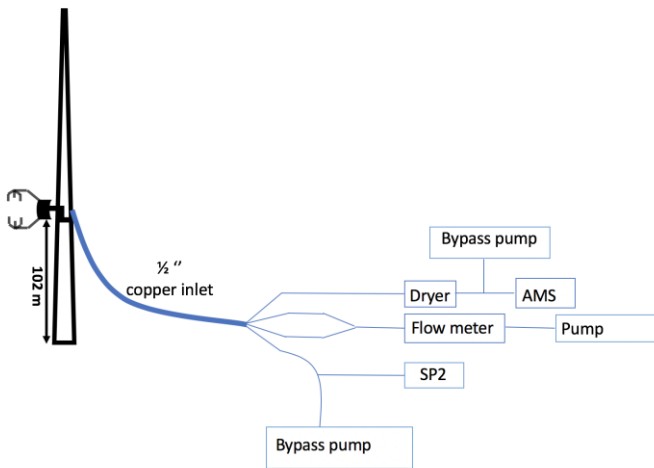


**Figure 1. Eddy covariance (EC) measurement setup on the IAP tower for BC flux measurements. The aerosol and BC were sampled from an inlet placed at 102 m above ground level. The sonic anemometer 0.75 m from the inlet measured the wind components and virtual temperature.**

**2.4 Black carbon flux calculations**

The EC method derives the net flux of BC through the horizontal plane at the height of the measurement from the correlation
between the fluctuations in the measured vertical wind speed (w) and BC concentrations, which can be related to the flux
between the surface and atmosphere (Emerson et al., 2018).

$$F_{BC} = \overline{w'C_{BC}'} \qquad\qquad (2)$$

Where the net BC flux ($F_{BC}$) is calculated as the product of the instantaneous fluctuation in vertical wind speed ($w'$) and in BC
concentration ($C_{BC}'$), summed over an averaging period of 30 minutes. The fluctuations are derived using Reynolds
decomposition, e.g. $w' = w - \overline{w}$, where the overbar indicates an arithmetic mean. In the atmosphere, such fluctuations are
driven by turbulence and have time scales ranging from milliseconds to a few hours. Therefore, the net correlations of these
fluctuations are summed over a timescale that is sufficient to capture the majority of fluxes, without non-stationary influences
from true changes in emission or meteorology.



We use the widely used open source software, EddyPro software version 6.2.1 (LI-COR Inc.) to calculate the fluxes as used with $CO_2$ and other greenhouse gas flux measurements. A time-lag occurs between wind and BC measurements from the

spatial separation of the sonic anemometer and the SP2. Time-lags can be determined for each 30 min period, but it can be difficult to quantify when the measurements have a low signal-to-noise and fluxes are associated with higher uncertainty (Langford et al., 2015). Here, time-lags are calculated using 24-hour periods of 5 Hz data, with application of covariance maximisation method (Nemitz et al., 2008; Su et al., 2004), giving a time-lag for each day. As these are found to be consistent, a mean constant time-lags of 13 (winter) and 14 seconds (summer) is used. For streamline corrections, the double rotation

method is used with the anemometer three velocity components (Wilczak et al., 2001). Finally, the fluxes were calculated using linear detrending method for each 30 minutes of averaging period, but only if the averaging period had missing data gaps that accounted for less than 10% of the data loss. Missing data are caused by either interruption in sampling (e.g. instrument maintenance, calibration) or problems with the logging computer (Section 2.2). The random uncertainty in the measurements is calculated following Finkelstein and Sims. (2001) approach.


## 2.5 Environmental and micrometeorological conditions

The vertical transport of the BC emitted depends on micrometeorology and environmental conditions. The friction velocity ($u_*$) is a measure of the efficiency of the vertical transport of momentum and thus also of the pollutants (Figure 2(a)). During

the early hours of the day, in the absence of solar heating, friction velocities are lowest, are more dependent on turbulence generated from wind shear. After sunrise additional heating (solar, and human activities) can enhance turbulence making it more buoyant. This increase occurs in both seasons. This effect is stronger in summer as the radiation is stronger and the stored heat is greater (Figure 2(b)). Furthermore, it is crucial to confirm that measurements are within the boundary layer so can be related to surface emissions. The mixed layer heights (MLH) are derived from ceilometer (Vaisala CL31, Finland)

measurements at the base of the tower using CABAM algorithm (Kotthaus and Grimmond, 2018). In both seasons the average MLH is greater than the measurement height (Figure 2(c)).

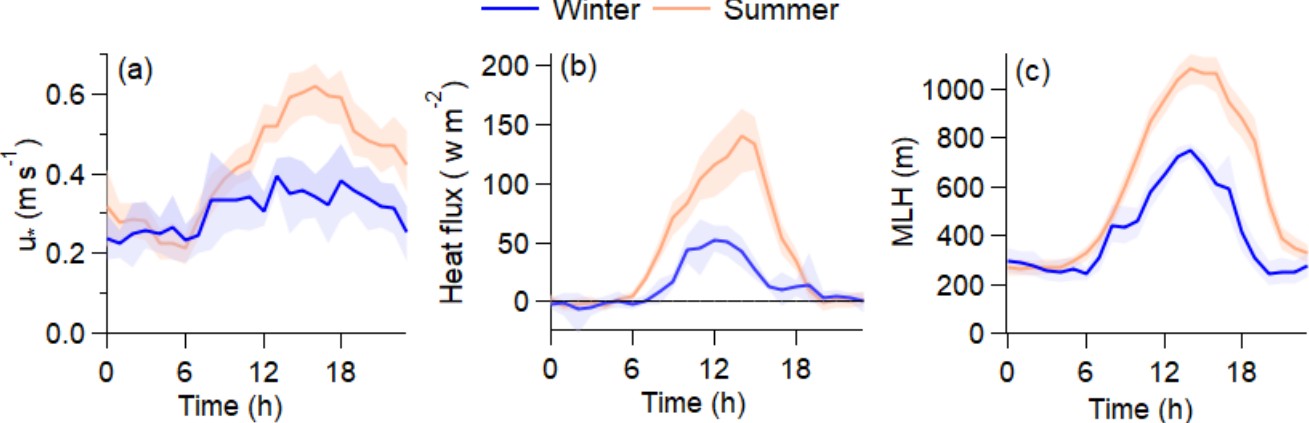

**Figure 2. Winter (blue) and summer (orange) diurnal pattern of: (a) friction velocity ($u_*$), (b) sensible heat flux and (c) mixed layer height (m). Shaded areas are the interquartile ranges, solid line the mean.**

## 3 Quality assurance and corrections


### 3.1 Stability, stationarity and storage corrections

Rural fluxes are often filtered to remove low friction velocity periods (e.g. $u_* < 0.2$ m s$^{-1}$) when vertical exchange is suppressed, and surface emissions may not reach the measurement height but may instead be advected (e.g Barr et al., 2013). For urban

flux measurements, given the large storage and anthropogenic heat fluxes the urban areas may remain unstable (e.g Kotthaus and Grimmond, 2012, 2014)) or not (Ward et al., 2013) making a single cut-off value of $u_*$ challenging because, unlike for rural $CO_2$ exchange, times during which fluxes should be independent of $u_*$ are difficult to predict. This is due to the complex boundary layer behaviour caused by the heterogeneity of the urban canopy, urban heat island effect and release of storage heat to the boundary layer. In this study, we applied a filtering threshold of $u_* < = 0.15$ m s$^{-1}$. The data loss from this is higher in

winter (21%) than summer (13%). Removing these data increases mean BC fluxes; the average mass and number fluxes increased by 10% during winter and in summer, the mass and number fluxes increased by 14% and 13 % respectively. Periods of lower turbulence typically occur at night when the MLH is also lower. In those conditions, without efficient transport of emissions to the measurement height, a build-up in concentration below this level may occur (i.e. positive storage flux). This build-up is typically released later when the boundary layer grows (i.e. negative storage flux). Here, a one single-point storage

flux correction is calculated as (Rannik and Vesala, 1999):

$$F_s = \frac{c(t+\Delta t) - c(t-\Delta t)}{2\Delta t} \times z \tag{3}$$





Here, $F_s$ is storage flux term, $c$ is the concentration, $z$ is the measurement height and $\Delta t = 30$ min. The storage correction was

considered when the surface emission required (e.g. diurnal cycles). For deriving correlations between co-emitted pollutants with the BC flux, storage correction is not applied to any of the metrics as this is not required and could potentially add additional error.

**3.2 Spectral analysis**


Spectral analysis is a useful approach of diagnosing the nature of turbulence captured by an EC system (Kaimal et al., 1972). Generally, power spectra of the measured scalar ($x$) are used to demonstrate an instruments response for capturing the range of turbulent fluctuations, but the poor BC count statistics (section 2.2) resulted in power spectra that resembled white noise (not shown). Instead, the covariance spectra (vertical wind and scalar, w'T and w'BC') are shown in Figure 3. To minimise

the effect of outliers in the spectra, a median is used (Järvi et al., 2014) with instantaneous fluxes less than 0 ng$^{-2}$ s$^{-1}$ (both seasons) and low frictional velocities ( less than 0.25 m s$^{-1}$ in winter and 0.38 m s$^{-1}$ in summer) removed. BC mass co-spectra follow sensible heat co-spectra until the attenuation due to noise at around a non-dimensional frequency of 1, suggesting that scalar transport occurs from energy transporting eddies. Additionally, the peak in the spectra occurs towards lower frequencies showing they are dominant for transporting the majority of the fluxes. The trend in high frequency at the inertial subrange

range follows the $f^{-4/3}$ response as predicted from Kolmogorov theory (Kaimal and Finnigan, 1994) (Figure 3, purple). The attenuation of the BC mass and consequently BC number (not shown here) occurs after 1. Given the majority of the flux is driven by much lower frequencies, a correction for high frequency loss is not applied.

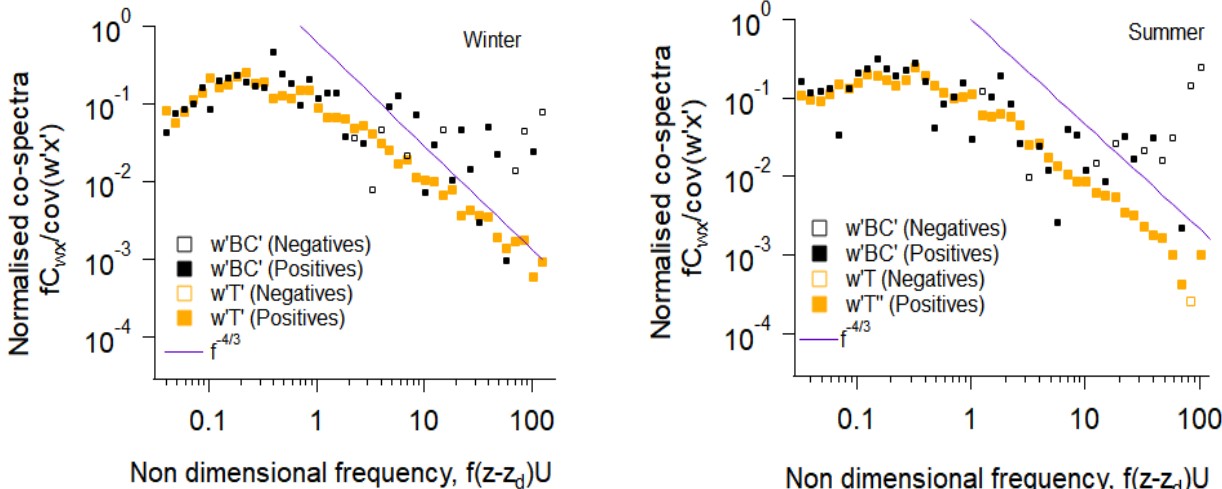

**Figure 3. Median of BC flux (black) and temperature (sensible heat) (yellow) co-spectrum multiplied by natural frequency (f) and normalised using total covariance against non-dimensional frequency. Data are binned into 44 logarithmically evenly-spaced bins, and co-spectra with total BC flux > 0 ng$^{-2}$ s$^{-1}$ and carefully selected frictional velocity for more turbulent periods (see section 3.2). A total of 223 (winter) and 170 (summer) cases are averaged.**



# 4 Results

## 4.1 Tower and ground comparison

The physical properties of BC in Beijing during this campaign for both seasons have been discussed by Yu et al. (2020) and Liu et al. (2019) based on the concurrent ground-level SP2 observations. The mass ratio of internally mixed non-BC material (the 'coating') to BC was quantified by Yu et al. (2020), which varied from 2 to 12 in the winter season and 2 to 3 in the summer season. The higher variability during the winter was related to combination of frequent occurrences of heavy haze episodes and presence of additional sources compared to the summer seasons. Liu et al. (2019) performed source apportionment analysis for the same ambient BC measurements and identified four different BC modes, which were then compared with AMS and SP-AMS measurement to relate each mode with their potential pollutant source. These four modes were identified using the $E_{sca}$ method described in Section 2.2, and the four modes include: thickly coated (coating diameter ($ct$) > 200 nm), small thinly coated ($ct$ < 50 nm and $D_c$ < 180 nm), moderately coated (50 nm > $ct$ < 200 nm) and large thinly coated ($ct$ < 50nm and $D_c$ > 180 nm). Their study also suggested that thinly coated particles were related to traffic-related activities and, thickly coated and large thinly coated particles were related to solid fuel burning. Since our study aimed to measure BC fluxes for particles characterised according to their sources, a similar analysis was performed for the tower-level measurements as well as ground level, to understand the suitability of tower level measurements for drawing similar source characterisations. Firstly, here we compare the concentrations of BC at both levels. Figure 4(a) shows the overlap periods of both measurements and Figure 4(b) shows correlation between the two sampling heights with an $R^2$ = 0.94 and 0.70 for winter and summer, respectively. The differences in the measurements may partially be due to line losses. Secondly, we characterised the mixing state of the particles for the tower level using the same scattering enhancement method as Liu et al. (2019), which is shown in Figure 5. The two chosen scenarios include polluted and clear periods to capture a variety of BC sources. Our analysis shows similar characteristics of the physical properties of BC as those observed from the ground measurements. Instead of four distinctive modes, we see some overlaps between those modes.



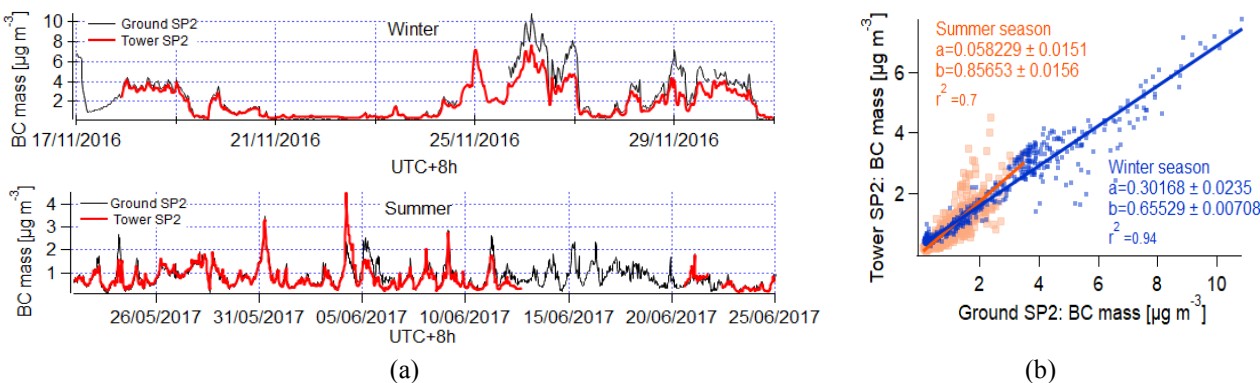

**Figure 4. (a) Comparisons of BC concentrations at 5 m (black) and 102 m (red) in winter and summer. (b) Correlation of BC concentrations measured at the two different levels for winter (blue) and summer (orange).**

This could be due to the use of an older generation of SP2 model, but also possibly because during the measurements the flux
instrument operated at relatively low laser power, either of which could have reduced the signal-to-noise of the scattering
channels. Therefore, we have grouped our measurements into two groups, *heavily coated* particles *($E_{sca}$>3)*, and *lightly coated*
particles *($E_{sca}$<3)*. The implication of *heavily coated* particles in this study refers to the combination of thickly coated and
moderately coated, i.e. solid fuel burning, while *lightly coated* in this study refers to the combination of small thinly coated
and large thinly coated particles defined and grouped by Liu et al. (2019) i.e. traffic particles.

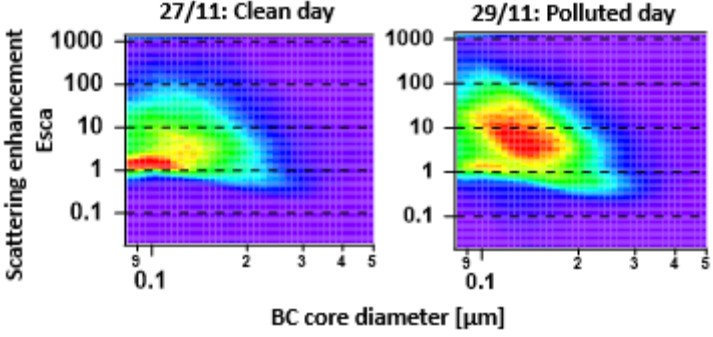


**Figure 5. BC particles physical properties on a clean and a polluted day. Scattering enhancement measured using the method discussed by Liu et al. (2019). Particle number density (colour) is > 70 % of the maxima in each panel the colour is set to red.**


**4.2 Black carbon fluxes**


**Figure 6. Time series of 30 minute averaged fluxes during winter (blue) and summer (orange) for mass and number fluxes with random error caused by precession limits of EC measurement system of this study (black bars).**

Figure 6 shows 30 minute fluxes of BC mass and number for both the winter and summer periods. The signal-to-noise of the mass fluxes in particular is low and fluxes vary throughout the measuring period, due to the variability in emissions in the flux footprint, e.g., in response to activities and time of the day. However, mainly positive fluxes were observed confirming a net emission of BC from the city. During the winter period, measurement coverage was lower (total number of 30 minutes averaging interval = 505) compared with the summer (total number of 30 min averaging interval = 976). Mean averaged mass fluxes (which associated standard error) of $5.49 \pm 0.49$ ng m$^{-2}$ s$^{-1}$ and $6.10 \pm 0.18$ ng m$^{-2}$ s$^{-1}$ were measured during the winter and summer, respectively. For the BC number flux, averages of $261.25 \pm 10.57$ cm$^{-2}$ s$^{-1}$ and $334.37 \pm 0.37$ cm$^{-2}$ s$^{-1}$ were measured. In order to remove any time-of-day biases associated with missing or filtered data, gaps in the data could be replaced with the corresponding average diurnal value. This delivers modified BC mass fluxes of 5.22 and 6.06 ng m$^{-2}$ s$^{-1}$ for winter and summer respectively and BC number fluxes of 252.41 and 334.40 cm$^{-2}$ s$^{-1}$). The uncertainty of the measurements is shown with black error bars in Figure 6. The calculation of such uncertainty was performed by calculating the random error (RE) of



each averaging period following the method discussed by Finkelstein and Sims. (2001). The mass flux is associated with a larger uncertainty than the number flux as the mass flux suffers from poor counting statistics since few particles with large BC mass make a large contribution to the total flux. It is important to note that aerosol flux measurements tend to have higher RE than gas flux measurements. While the point-by-point data is of low precision, averaging of the fluxes according to either diurnal cycle or wind sector can deliver more meaningful statistics, and this is discussed in the following section. Additionally,

the summary of BC concentrations and fluxes is also provided in Table 1.

| Concentrations | BC mass (µg m$^{-3}$) | | BC number (cm$^{-3}$) | |
|---|---|---|---|---|
| | Winter | Summer | Winter | Summer |
| Mean | 1.90 | 0.70 | 677.70 | 330.00 |
| Median | 1.32 | 0.53 | 495.92 | 268.57 |
| Standard deviation | 1.54 | 0.51 | 558.33 | 239.90 |
| Percentile | | | | |
| 5th | 0.37 | 0.14 | 126.85 | 77.77 |
| 95th | 4.70 | 1.56 | 1656.40 | 751.70 |
| Number of observations | 716 | 1428 | 716 | 1428 |
| Fluxes | BC mass (ng m$^{-2}$ s$^{-1}$) | | BC number (cm$^{-2}$ s$^{-1}$) | |
| | Winter | Summer | Winter | Summer |
| Mean | 5.49 | 6.10 | 261.25 | 334.37 |
| Median | 4.13 | 3.58 | 203.71 | 229.01 |
| Standard deviation | 14.65 | 9.07 | 439.50 | 376.85 |
| Percentile | | | | |
| 5th | -13.87 | -3.45 | 218.77 | -43.66 |
| 95th | 29.13 | 22.78 | 1040.04 | 1091.60 |
| Number of observations | 505 | 976 | 505 | 976 |

**Table 1. Concentration and flux statistics of BC measurements at the IAP tower. The 5 Hz concentration measurement were re-sampled at 30 minutes period before preforming statistical calculations.**


**4.3 Local black carbon characterisation**





The flux footprint represents the local area that contributes to the measured flux. The shape and size of the flux footprint is related to the sampling height, wind speed and direction, atmospheric stability, surface roughness, turbulence and the boundary layer height. Figure 7 depicts the extent of the average flux footprint for the winter and summer measurement periods, respectively. These footprints were generated using the method presented by Squires et al. (2020), using the footprint model and theory discussed by Kljun et al. (2004) and Metzger et al. (2012). However, the total numbers of individual footprints used to generate average campaign footprints are slightly different to those presented by Squires et al. (2020) owing to slight differences in the respective instrument uptimes. The footprints were superimposed onto a map of Beijing at a grid resolution of $10 \times 10$ m$^2$. The initial comparison between the seasons shows that the averaged sizes of the footprints were similar. Here, the counter circles represent 30%, 60% and 90% cumulative contribution to the measured averaged fluxes from those regions. During both seasons the majority (up to 90%) of contributions reached to about 2 km from the IAP tower. However, within that range maxima of the contribution occurred at a distance of 0.26 km. The major difference between the two seasons was the change in the dominant wind directions. During winter, the wind direction was predominantly from the north-west and hence measurements are likely to have captured emissions from Beitucheng West Road in the NW direction. During summer, winds were predominantly from the north-east direction capturing emissions from a major Jingzang Expressway. Since this analysis indicates the presence of traffic-related emissions, we have shown diurnal trends as well as their spatial intensity according to their wind speed and direction in the following section.

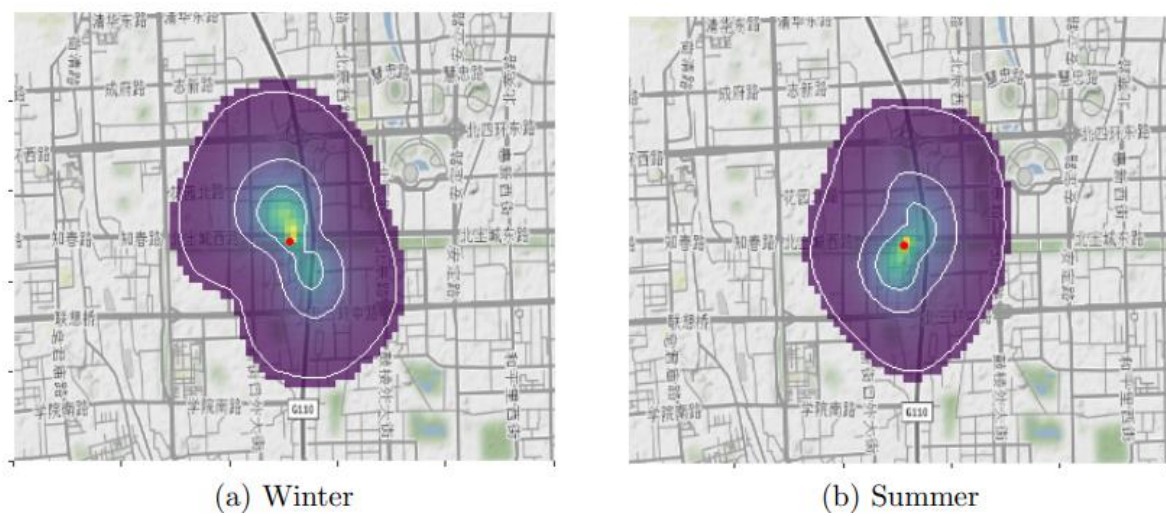

(a) Winter       (b) Summer

**Figure 7. Total average mass flux footprints for winter and summer seasons. The red spot on each plot is the location of the IAP tower, and surrounding colours represents the magnitude of the intensity of the measured emission. The resolution of footprint is 100 m$^2$, the outermost white line represents the 90% cumulative spatial contribution of the total flux and the middle and inner most lines representing 60% and 30% contribution. Map tile sets are © Stamen Design, under a Creative Commons Attribution (CC BY 3.0) license.**

.





## 4.4 Diurnal and wind sector trends

Figure 7 shows the diurnal cycles for mass and number fluxes, for winter (blue) and summer (orange) and error bars represents
the associated averaged random errors (e.g. $\overline{RE_{hour}}$). These were calculated using ensample average of RE for each diurnal
hour as:

$$\overline{RE_{hour}} = \frac{1}{N}\sqrt{\sum_{i=1}^{N} RE_i^2},$$    (4)

where $N$ is the number of 30 min averaging periods that entered into generating the average value for each time of the day.

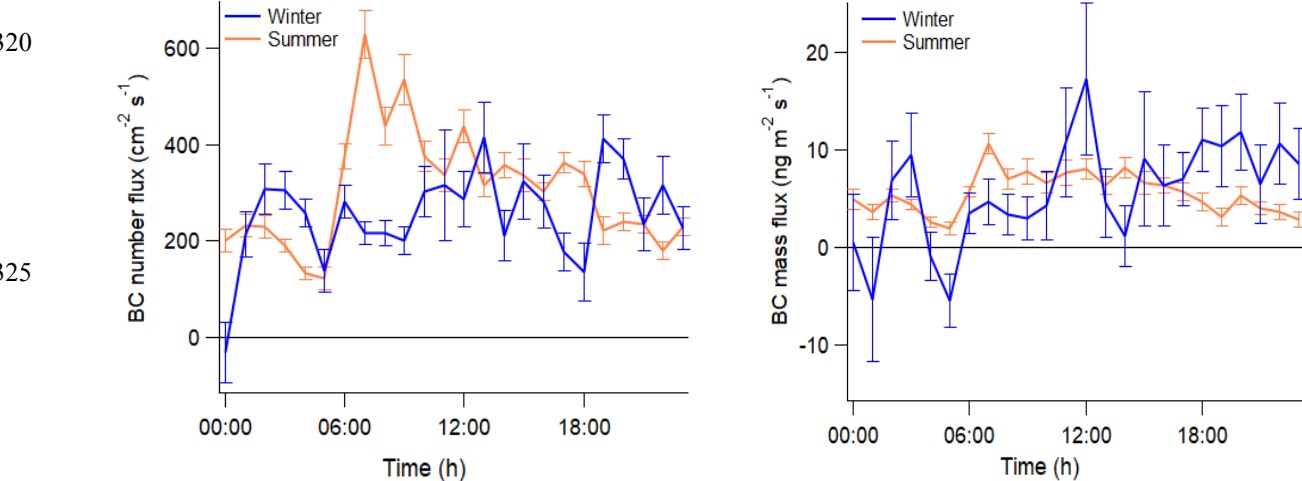

**Figure 8. Diurnal cycles for (a) mass and (b) number flux in summer and winter with error bars (standard errors, from the calculated precision). For this analysis storage corrected fluxes are used to help understand ground based emissions.**

During winter, for both mass and number flux, there were no clear diurnal cycles. In summer there was a clear peak at 7 am,
which could be related to traffic-related activities, but there was no distinctive peak for the evening rush hour. However, the
average flux values between the two seasons were similar, indicating a similar source between seasons. For both seasons, BC
diurnals showed similar patterns to that of CO and $NO_x$ fluxes (Squires et al., 2020), suggesting that traffic emissions are the
dominant source of BC in this region of Beijing. Therefore, we additionally averaged BC fluxes according to wind sector, and
generated polar plots using *OpenAir* (Carslaw and Ropkins, 2012) for mass fluxes, as shown in Figure 9. This analysis clearly
shows that fluxes were largest in the NE direction during both seasons. This is crucial as, during the winter, the NE is not a
prominent wind direction as we discussed in the footprint analysis section. The analysis would suggest that the strongest
sources lie in the NE and E directions, which is consistent with the presence of a major traffic junction. During the summer,
the indications that the major source is traffic related (such as the diurnal profile) are generally clearer, due to better coverage
and signal-to-noise, turbulence and favourable wind direction. The quantification of traffic emissions is discussed in the
following section.


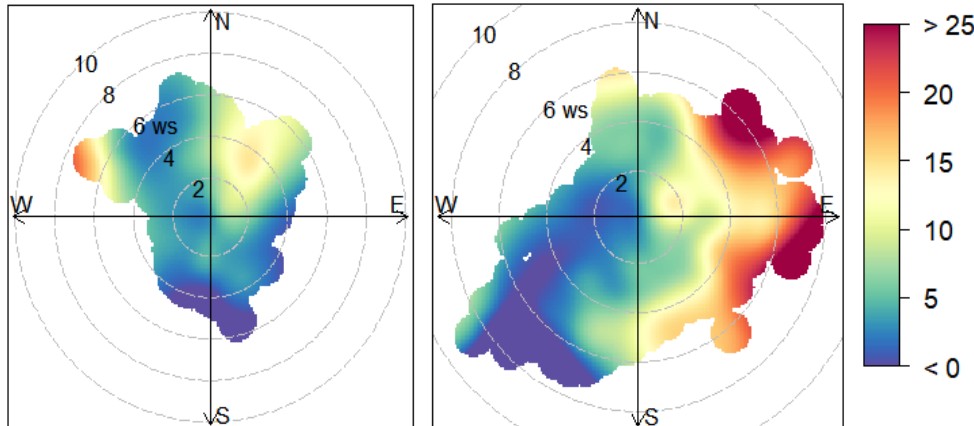

**Figure 9. BC mass flux distribution by wind direction and wind speed (a) winter and (b) summer analysis done using *OpenAir* software (Carslaw and Ropkins, 2012).**

**4.5 Flux for characterised black carbon particles.**

In Section 4.1 the source apportionment of BC particles, according to their coating content was discussed using the threshold value of $E_{sca}=3$, particles above this threshold are defined as *highly coated* and the rest of the particles are *lightly coated*. *Lightly coated* particles were considered from traffic related emissions and *heavily coated* particles from solid fuel burning. Additionally, Liu et al. (2019) suggested particles with $D_c > 180$ nm can be associated with coal combustion. Therefore, in this analysis, we firstly separated BC mass and BC number concentrations using threshold value of $E_{sca}=3$ and defined this

characteristic as *coating classification*. The average mass and number concentration of *lightly coated* and *heavily coated* particles are shown in Table 2. Almost a quarter of the BC mass concentration (and slightly less than a quarter if we consider number concentrations) measured at IAP were from solid fuel burning (*highly coated*) and the rest of the particles were from traffic emissions (*lightly coated*). In order to quantify what proportion of these particles was emitted locally, fluxes of each individual categories were also calculated, and their average values are also summarised in Table 2. We can see that the

contribution of BC particles emitted from solid fuel burning was almost negligible for both BC mass and number fluxes, with traffic emissions accounting for 92.3% of both fluxes.

    Secondly, since BC particles emitted from coal combustion are likely to have $D_c > 180$ nm, we used a threshold of $D_c = 180$ *nm* to perform the *size classification*. It is important to note that there were fewer particles above this range, yet they contributed significantly to the concentration in mass space. Therefore, this analysis suffers from poor counting statistics and we cannot

reliably quantify emissions for particles with $D_c > 180$ nm. However, the general trend for both BC mass and BC number



quantity in Table 2 for the *size classification* shows that the average fraction of particles $D_c > 180$ nm is higher for concentration compared to their fraction in total flux measurements, indicating the majority of these particles are from outside of the flux footprint.

| | | Concentrations [%] | Fluxes [%] |
|---|---|---|---|
| | | *coating classification* | |
| BC mass | Highly coated (Traffic) | 23.2 | 7.7 |
| | Lightly coated (Solid fuel) | 76.8 | 92.3 |
| | | | |
| BC Number | Highly coated (Traffic) | 18.8 | 7.7 |
| | Lightly coated (Solid fuel) | 81.2 | 92.3 |
| | | *size classification* | |
| BC mass | $D_c > 180$nm | 63.8 | 58.3 |
| | $D_c < 180$nm | 36.2 | 41.7 |
| | | | |
| BC Number | $D_c > 180$nm | 15.6 | 7.5 |
| | $D_c < 180$nm | 84.4 | 92.5 |

**Table 2. Percentage values of concentrations and fluxes for *coating classification* and *size classification*. For *coating classification*, the total particles were separated according to coating contents using the threshold of $E_{sca}=3$ for both BC mass and number. For *size classification*, the total particles were separated at $D_c=180$ nm, for BC mass and number.**

**5 Discussion**

**5.1 Comparison with $NO_x$ and CO emissions**

On the assumption that CO and $NO_x$ are co-emitted with BC by combustion sources we have investigated the correlations of

their measured fluxes with BC using orthogonal distance regression. These are shown in Figure 10(a) and 10(b), for CO and $NO_x$, respectively, and the gradients in each plot correspond to the emission flux ratios. The comparison with CO fluxes shows BC/CO mass ratios of 0.0007 and 0.0011 for winter and summer, respectively. During the summer this ratio is 37% larger than in winter. Our study has shown similar emissions of total averaged BC in the two seasons; however, this is not the case for



CO. During winter, cold-start condition may explain the increase in CO emission which is further discussed by (Squires et al.,

2020). The comparison with $NO_x$ fluxes shows $BC/NO_x$ ratios are the same between seasons, 0.0052 and 0.0056 for winter and summer, respectively. The relationship in winter shows more scatter due to poor signal-to-noise. Nevertheless, the consistency of the $NO_x$ slopes is highly encouraging as the identical relationship is not only consistent with the emissions being from a singular dominant source, i.e. vehicles, but also provides confidence in the measurements, given that a physically different instrument was used in the two seasons.

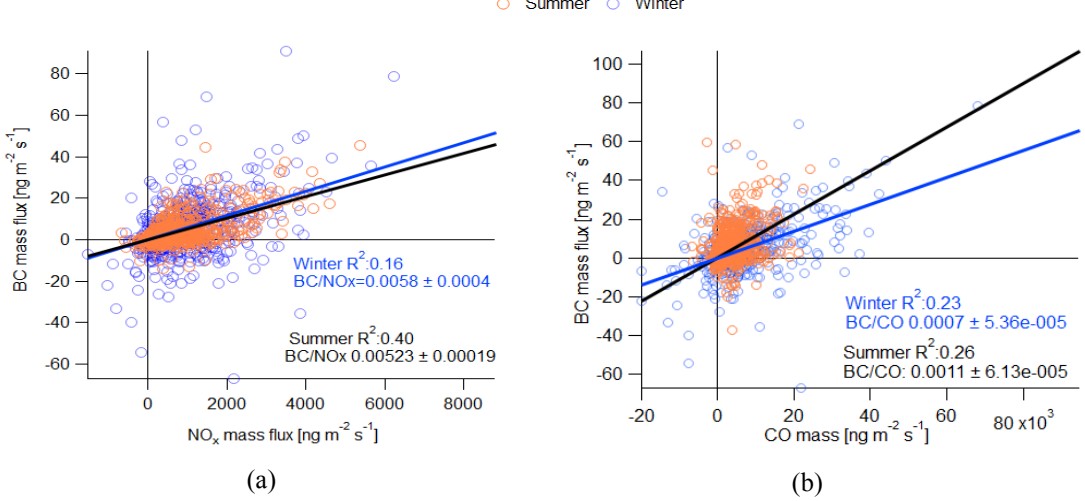

**Figure 10. BC mass flux with (a) $NO_x$ flux (Squires et al., 2020) and (b) CO flux (Squires et al., 2020) for winter and summer with calculated $BC/NO_x$ and $BC/CO$ ratios, $NO_x$ and CO fluxes converted to the same units as our measurements.**

Following a similar vehicular European regulation system for emission controls, China has started to adopt emission regulation targets. The limits imposed cover the major pollutants CO, $NO_x$, NMHC, $HC+NO_x$ and PM (which refers to $PM_{2.5}$) for both Light-Duty Gasoline Vehicles (LDGVs) and Heavy-Duty Diesel Engines (HDDEs), as summarised by Wu et al. (2017). However, to the best of our knowledge, reliable values of BC fraction within $PM_{2.5}$ are not available in the literature for the China's regulation standards. Therefore, we used the average $BC/PM_{2.5}$ ratios for gasoline engines from the European

guideline of emissions controls as stated in Table 3.2.1 of Ntziachristos et al. (2017) which are 0.12 and 0.57 for gasoline and diesel engines, respectively. Using these ratios, we have estimated $BC/NO_x$ and $BC/CO$ ratios implied with each China emission's standards for both LDGVs and HDDEs, as shown in Table 3. It is important to note that $PM_{2.5}$ controls for LGDVs were only introduced recently from 2017/01/01 as part of China 5 controls. If we compare our measured emission ratios as shown in Figure 10 with the values in Tables 3, we find that the measured values are in a similar range as the ratio of China 5

limit levels for gasoline emissions, noting that diesel vehicles are banned from central Beijing during the daytime and are therefore not expected to contribute. Making the implicit assumption that the legal limits are a fair reflection of actual emissions, this is in a good agreement.





|  | Implementation date | BC/NO$_x$ | BC/CO |
|---|---|---|---|
| **Gasoline emissions standards** | | | |
| **China 1** | 2000/7/1 | - | - |
| **China 2** | 2005/7/1 | - | - |
| **China 3** | 2008/7/1 | - | - |
| **China 4** | 2011/7/1 | - | - |
| **China 5** | 2017/1/1 | 0.009 | 0.00054 |
| **China 6a** | TBD | 0.009 | 0.00108 |
| **China 6b** | TBD | 0.01031 | 0.0007 |
| **Diesel emission standards** | | | |
| **China I** | 2001/9/1 | 0.026 | 0.046 |
| **China II** | 2004/9/1 | 0.012 | 0.021 |
| **China III** | 2008/1/1 | 0.012/0.018 | 0.027/0/017 |
| **China IV** | 2015/1/1 | 0.0032/0.005 | 0.008/0.004 |
| **China V** | 2017/1/1 | 0.0057/0/009 | 0.008/0.004 |

**Table 3 Estimated ratio of BC/NO$_x$ and BC/CO for gasoline engines and diesel engines. An overview of on-road vehicular emissions and their control in China by Wu et al. (2017) summarised NO$_x$, CO and PM$_{2.5}$ emission regulation targets in China. The BC/ PM$_{2.5}$ fractions for gasoline were taken from European guideline of emissions controls, as such fraction is not available for China. Since PM$_{2.5}$ controls for LDGVs were only introduced recently from China 5 controls, our estimates could not be performed for previous standards.**

## 5.2 Emission inventory

Here we compare our measurements with the Multi-resolution Emission Inventory for China (MEIC; available at http://www.meicmodel.org/), developed at Tsinghua University (e.g. (Zhang et al., 2015)). The inventory uses spatial proxies (e.g. population and energy consumption statistics) to downscale emissions from national and provincial scale to finer resolution (Biggart et al., 2020; Qi et al., 2017). In this study, BC emissions for year 2013 at a resolution of 3 × 3 km$^2$ were used, which was the most recent version available at the time of this study. The inventory emission values were extracted for the area covered by the averaged flux footprint (see Section 4.3) following the same approach as Squires et al. (2020).

Figure 11 shows a comparison of the diurnal cycles between measured fluxes and the MEIC 2013 emission inventory for (a) winter and (b) summer, where the diurnal cycles for each sector (transport, industry, agriculture, residential and power) were generated from the MEIC 2013 estimates.



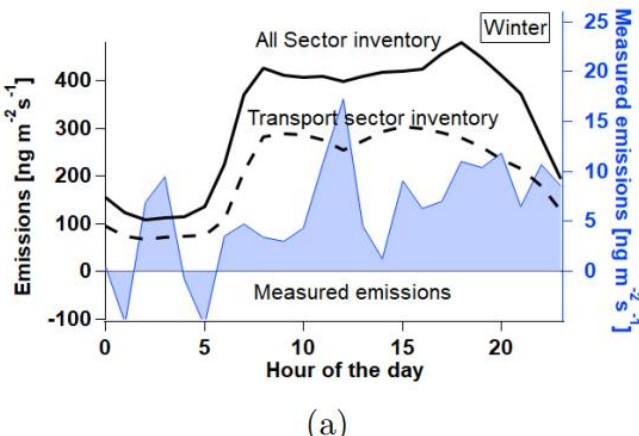
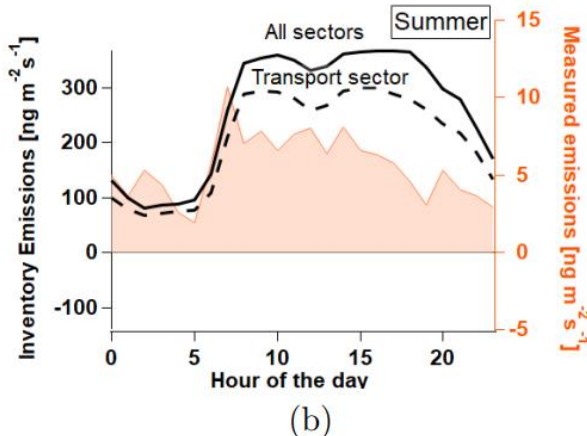

**Figure 11. Comparison between measured emissions and the MEIC 2013 emission inventory for (a) winter and (b) summer for flux footprint area (section 4.3). Diurnal variation from the MEIC estimates are used for the diurnal cycles for emissions of all sectors (sum of emissions from transport, industry, residential and power) and the transport sector in isolation.**

The inventory has a broadly similar temporal pattern to the observed fluxes but overestimates the total BC emissions from the

flux footprint by a factor of 59 and 47during the winter and summer periods, respectively. Since the transport sector contributes 63 % and 80 % of the overall BC emissions in the MEIC inventory, the average emissions of the transport sector in isolation were also compared with the measured flux. This reduced the discrepancy to within a factor of 38 for both winter and summer periods.

Reduction in emissions in Beijing from 2013 to 2017 have been analysed by Cheng et al. (2019) using the MEIC model in combination with a local bottom-up emission inventory. This study concludes that the reduction in $PM_{2.5}$ emissions from the transport sector is negligible, as since 2013 there has been an increase in stringent traffic management and fuel control policies, but vehicle densities have increased, keeping emissions from the transport sector steady. This is also discussed on a national level for BC emissions from the transport sector by Zheng et al. (2018) and concluded emissions from the transport sectors

remained steady between 2010 and 2017. Therefore, such large discrepancy is unlikely to be due to changes in actual emissions between 2013 and 2017. We also compared BC/CO and BC/$NO_x$ ratios for the transport emission data in MEIC, which were 0.0066, 0.0423 and 0.0072, 0.0425 for the winter and summer seasons, respectively. Compared to the observed ratios, these are factors of 9.4 and 6.5 too high for the ratios with CO in winter and summer respectively, and 7.3 and 8.1 for the ratios with $NO_x$. This implies that the BC emissions from traffic are overestimated in MEIC, although it should be noted

that the emissions of $NO_x$ and CO from the inventory are also high compared with measured fluxes, as reported by Squires et al. (2020). The overestimation of emissions in this region may reflect the suitability of the proxies used in downscaling the emissions from a regional, provincial resolution to the 3-km scale used here, as noted by Zheng et al. (2017). However, the



overestimation of the ratios highlights that it is likely that BC emissions are overestimated in MEIC, at least for the region considered here. The diurnal pattern of emissions, in contrast, are in relatively good agreement.


**6 Conclusion**

Local emissions of BC fluxes were measured using the eddy covariance method in Beijing as part of the APHH project during the winter of 2016 and summer of 2017 from the IAP tower, the first application of this approach to an urban environment.

During both seasons, average BC mass flux and number fluxes remained stable. In the case of BC mass fluxes, average values of $5.5 \pm 0.49$ ng m$^{-2}$ s$^{-1}$ and $6.1 \pm 0.18$ ng m$^{-2}$ s$^{-1}$ were measured during the winter and summer seasons, respectively. In the case of BC number fluxes, average values of $261.3 \pm 10.57$ cm$^{-2}$ s$^{-1}$ and $334.4 \pm 0.37$ cm$^{-2}$ s$^{-1}$ were measured for the winter and summer seasons, respectively. The similarity in the magnitude of emissions between seasons suggests that there was no major additional source in winter associated with heating, consistent with the use of district heating in central Beijing. Flux

footprints (up to 2 km from the IAP tower in both seasons) and wind sector-based analysis showed emission sources are strongest to the NE and E of the IAP tower, where a major road is situated, confirming the presence of traffic emissions. The contribution of traffic emission was quantified by classifying total BC particles according to coating thickness during the winter season, following the previous observation that traffic emits smaller, more lightly coated particles. The analysis indicated traffic emissions were approximately 92% of total measured flux for both BC mass and BC number fluxes, and the

rest was possibly associated with solid fuel burning. In terms of the overall concentrations, the heavily coated particles represented 23% and 19% of the BC mass and number concentrations respectively. The higher presence of the heavily coated particles in concentration space, but not local fluxes suggests advection of such source from outside the footprint and is indicative of regional pollution. The measurements were also compared with NO$_x$ and CO fluxes and the corresponding emission ratios BC/CO and BC/NO$_x$ were inferred. The BC/NO$_x$ ratio was found to be consistent between seasons and both

ratios were in general agreement with ratios implied by the China emission standards. Finally, the measured emissions were compared with the MEIC 2013 inventory. The magnitude of the transport sector emissions in the inventory alone was significantly larger (by over an order of magnitude) than our measurements. Such a discrepancy cannot solely be explained by changes in emissions since 2013, indicating that there are other inaccuracies, likely in the proxies used to downscale the emissions to an urban landscape as noted by Zheng et al. (2017). The shape of the diurnal variation in the inventory was in

better agreement, however, the BC/CO and BC/NO$_x$ ratios from the MEIC inventory were substantially larger than the measured ratios, which could indicate that an inaccuracy in the assumed BC emission profile is at least partially responsible for the disagreement. Indicating that BC emissions in the inventory are overestimated, at least in the urban area of Beijing.

**Data Availability**

Processed data is available on the APHH-Beijing project database on the Centre for Environmental Data Analysis (http://data.ceda.ac.uk/badc/aphh/data/beijing/). Raw data is available on request.



**Acknowledgements**

Rutambhara Joshi's PhD was supported by National Centre of Atmxospheric Science (NCAS). The field measurements were
supported by the the Newton Fund, administered by the UK Natural Environment Research Council (NERC) through the AIR-
POLL and AIRPRO projects of the Air Pollution and Human Health in a Chinese Megacity (APHH-Beijing) programme (grant
references NE/N006992/1, NE/N006976/1, NE/N006917/1, NE/N007123/1 and NE/N00700X/1).

**Author contributions**


RJ made the measurements, performed data analysis and wrote the paper with support from JA, HC and DL. DL helped with
developing toolkit for pre-processing of SP2 flux data and provided guidance with source apportionment of work. NM installed
and maintained the tower EC measurement setup. EN and BL helped RJ with flux calculations and use of EddyPro software.
FS and JL provided flux footprint calculations, with corresponding emission inventory data. YW, XP and PF allowed to use
their SP2 instrument during summer season. SK and SG provided the mixed layer height data. QZ and RW provided MEIC
inventory data and OW helped with the interpretation of the inventory values. MF provided technical support for maintenance
of the SP2 instruments. HC and JA are PhD supervisors of RJ. All the authors read and improved the manuscript.

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
