# Peer review of "Direct measurements of black carbon fluxes in central Beijing using the eddy-covariance method."

_Atmospheric Chemistry and Physics, 2020_

## Referee Comment (RC1) · Anonymous Referee #1 · 27 Aug 2020

The authors present the first application of the eddy covariance method to measure black carbon (BC) fluxes in an urban environment. They report turbulent BC number and mass fluxes in the winter and summer seasons, and identify traffic as the dominant source during both seasons at their site in central Beijing. The authors also compare observed BC fluxes as well as BC/NOx and BC/CO emission ratios with a 2013 emission inventory (MEIC) and show similar temporal patterns but large discrepancies in emission strength. The manuscript is well written, the method and its limitations are thoroughly discussed, and the presented results are original. The manuscript is a very important contribution to the field. Therefore, it should be published in ACP after minor revisions:

Specific comments: The title of Section 3.1 is "Stability, stationarity and storage corrections" but this section only deals with the storage correction term. I suggest to change the section title to "Storage correction". Stationarity is not mentioned at all. I would encourage the authors to add a brief paragraph on stationarity.

I very much appreciate the spectral analysis presented in section 3.2. The authors state that "BC mass co-spectra follow sensible heat co-spectra until the attenuation due to noise at around a non-dimensional frequency of 1, suggesting that scalar transport occurs from energy transporting eddies." I am not sure if I understand the second part of this statement correctly. There is spectral similarity up to a non-dimensional frequency of 1 but it's important to point out that the spectral peak and the major fraction of the cospectral density is found at lower frequencies in order to suggest that scalar transport is mainly with energy-transporting eddies. The spectral peak is mentioned only after this conclusion. Also, in lines 219/220, it should be added that the -4/3 slope in the inertial subrange is evident for the sensible heat flux.

In section 4.1, the authors introduce four different BC modes identified by Liu et al. (2019) who "suggested that thinly coated particles were related to traffic-related activities and, thickly coated and large thinly coated particles were related to solid fuel burning". The authors continue to introduce two new groups, heavily coated referring to thickly coated and moderately coated particles from solid fuel burning, and lightly coated referring to small thinly coated and large thinly coated from traffic particles. There is a contradiction regarding large thinly coated particles, related to solid fuel burning in line 238 but to traffic in line 254. Please clarify!

In the top panel of Figure 4a), there seems to be a systematic change in the difference between tower and ground-based measurements in the winter campaign from 25 November. Could you briefly discuss what might have happened? Also, in Figure 4b) please introduce the meaning of a and b as offset and slope, or remove.

Section 4.2 presents very interesting BC flux data, and I agree that while there is some

uncertainty due to poor counting statistics, averaging can yield meaningful statistics. What is striking in Figure 6 is the much larger amplitude of net fluxes (both number and mass) in winter compared to summer. I recommend adding a brief passage that discusses the fraction of net deposition fluxes during the winter and summer campaigns, and the stronger net emission flux peaks in winter compared to summer. In Figure 7, please add a 1 km scale and indicate the two major roads mentioned in the text to make the map more meaningful.

In Section 4.4, line 335, the authors state that "average flux values between the two seasons were similar, indicating a similar source between seasons". In my opinion, the second part is an overstatement.

Section 4.5 is very interesting and original but I was left wondering whether or not seasonal differences occurred. I would strongly recommend to present Table 2 for the winter and summer seasons separately. This could also add to the discussion of a similar dominant source in the winter and summer seasons, which is also mentioned in the Conclusions in line 449. In Table 2, use the term "heavily coated" instead of "highly coated" for consistency, and in the second column, exchange the associated sources given in parentheses, i.e. heavily coated (solid fuel) and lightly coated (traffic).

Technical comments: Put table captions above tables. l. 39: Remove "." between "diseases" and "(". l. 39: "exposure limit" instead of "exposure limits". l. 43: Check units of "a 1 $\mu$m3 increase". l. 45: "with warming potential to the climate second only to CO2" - the term "warming potential" is confusing. I suggest using "radiative forcing" instead. l. 97: Remove "W." from reference Bond and Bergstrom (2006). l. 106: Remove "." between "Kondo" and "(". l. 164: "constant time-lag" instead of "constant time-lags". l. 169: Remove "." between "Sims" and "(". l. 212: "an instruments' response" instead of "an instruments response". l. 263: What do you mean by "precession limits"? l. 269: "with associated standard error" instead of "which associated standard error". l. 273: Remove ")" before ". The uncertainty..." l. 275: Remove "." between "Sims" and "(". l. 284: "before performing" instead of "before preforming". l. 314: "error bars represent"

[Figure]

instead of "error bars represents". l. 315: "ensemble average" instead of "ensample average". l. 318: I recommend "for each hour of the day" instead of "for each time of the day". l. 330: Figure caption should read "Diurnal cycles for (a) number and (b) mass flux...". l. 346, Figure 9: What are the units of the color scale? l. 357: "heavily coated" instead of "highly coated". l. 360: Instead of "almost negligible", I recommend "smaller than 8 %". In my opinion, a contribution of 7.7 % is not negligible in this context. l. 379/380: "discussed by Squires et al. (2020)" instead of "discussed by (Squires et al., 2020)". l. 380: Values for BC/NOx ratios given in this line are not in agreement with values presented in Figure 10a). l. 388: Remove "the" after "in the literature for". Table 3: On p. 18, Table 3, I recommend adding the observed values for direct comparison. l. 398: In Table 3, in the third row, China V change "0.0057/0/009" to "0.0057/0.009", and in the last row, China III change "0.027/0/017" to "0.027/0.017" l. 408: Remove parentheses around "Zhang et al., 2015". l. 410: "BC emissions for the year 2013" instead of "BC emissions for year 2013". l. 436: "may reflect the limited suitability" instead of "may reflect the suitability". l. 459: Add "." between "inferred" and "The BC/NOx". l. 467: "This indicates" instead of "Indicating". l. 474: Remove "x" in "Atmxospheric".
* * *

---

## Referee Comment (RC2) · Anonymous Referee #2 · 26 Sep 2020

Joshi et al. present a truly excellent example of how atmospheric chemical measurements can be used to directly improve models and provide new insight into underlying processes. The paper is well written, the data are robust, and the analysis is strong. This paper adds substantively to the literature. The flux uncertainty estimates, footprint analysis and tables of statistics are useful and well done. I recommend publication following minor revisions. Comments are below.

Comments:

1. In the Methods, please be clear on the actual length of the inlet line: it's on a 102 m tower, but presumably longer? Figure 1 would be much improved with line lengths and

flow rates, making the field setup more reproducible. Please provide estimates on particle losses from the inlet lines in the general size range. While those losses might not influence the calculated fluxes, size-dependent losses can cause bias in total particle mass or number fluxes – and some underestimates of emissions (or deposition).

2. A technical point, but Figure 2 woud be easier to interpret if the x-axis ran from 0-24 hours (rather than ending at 18).

3. Figure 3 makes me worry about signal loss due to attenuation. While the authors write that "Given the majority of the flux is driven by much lower frequencies, a correction for high frequency loss is not applied." However, this fact is not clear from Figure 3. Please specify what percentage of the flux is driven by lower frequencies (and how that is defined), and what the influence of a correction would be. That is, would correcting the flux in Figure 3 for attenuation cause a 5% change (in which case, definitely minor), or a 30% change (in which case the corrections should be applied)?

4. The significant figures in Figure 4(b) and 10 are excessive. Please correct to something more reasonable. Also, it is not clear if 'a' is the slope or intercept in Figure 4b? Please specify on the figure what 'a' and 'b' refer to.

---

## Author Comment (AC1) · 6 Nov 2020

"Direct measurements of black carbon fluxes in central Beijing using the eddy-covariance method" by Joshi et al.,2020, submitted to ACP

General response:
We thank both referees for their comments and helping us improve this manuscript. Both referees states that this work is useful contribution to the field and provided suggestion for further improvements. We provide response and correction to each comment here.

**Anonymous Referee #1

**The authors present the first application of the eddy covariance method to measure black carbon (BC) fluxes in an urban environment. They report turbulent BC number and mass fluxes in the winter and summer seasons, and identify traffic as the dominant source during both seasons at their site in central Beijing. The authors also compare observed BC fluxes as well as BC/NOx and BC/CO emission ratios with a 2013 emission inventory (MEIC) and show similar temporal patterns but large discrepancies in emission strength. The manuscript is well written, the method and its limitations are thoroughly discussed, and the presented results are original. The manuscript is a very important contribution to the field. Therefore, it should be published in ACP after minor revisions:**

paper Specific comments:

1) The title of Section 3.1 is "Stability, stationarity and storage corrections" but this section only deals with the storage correction term. I suggest to change the section title to "Storage correction". Stationarity is not mentioned at all. I would encourage the authors to add a brief paragraph on stationarity.

*Response*

*We have added brief paragraph on stationarity, however we haven't changed the name of the section as now all the corrections are discussed and explained whether their application was required or not for this study.*

*Correction*

*Line 209-214: EC method assumes steady state conditions, which is often not satisfied due to change in weather patterns and meteorological variables with time of the day. The stationarity test described by Foken and Wichura. (1996) is often performed to examine whether a flux is statistically invariant over the averaging period. However, this test is unreliable for small fluxes with large uncertainties (e.g Nemitz et al., 2018)), therefore this filter is not applied in this study to avoid flagging of averaging periods as non-stationary irrespective of meteorological conditions.*

2) I very much appreciate the spectral analysis presented in section 3.2. The authors state that "BC mass co-spectra follow sensible heat co-spectra until the attenuation due to noise at around a non-dimensional frequency of 1, suggesting that scalar transport occurs from energy transporting eddies." I am not sure if I understand the second part of this statement correctly. There is spectral similarity up to a non-dimensional frequency of 1 but it's important to point out that the spectral peak and the major fraction of the cospectral density is found at lower frequencies in order to

suggest that scalar transport is mainly with energy-transporting eddies. The spectral peak is mentioned only after this conclusion. Also, in lines 219/220, it should be added that the -4/3 slope in the inertial subrange is evident for the sensible heat flux.

*Response*
*We tried to explain this section better by adding further explanation.*

*Correction*

*Line 228-232: BC mass co-spectra show the same co-spectral peak and follow sensible heat co-spectra up to a non-dimensional frequency of 5, at which the effect of noise becomes visible. This covers the majority of the flux contribution which occurs from lower frequencies. The trend in high frequency at the inertial subrange range follows the $f^{4/3}$ response as predicted from Kolmogorov theory (Kaimal and Finnigan, 1994) (Figure 3, purple), which is evident for sensible heat flux.*

3) In section 4.1, the authors introduce four different BC modes identified by Liu et al. (2019) who "suggested that thinly coated particles were related to traffic-related activities and, thickly coated and large thinly coated particles were related to solid fuel burning". The authors continue to introduce two new groups, heavily coated referring to thickly coated and moderately coated particles from solid fuel burning, and lightly coated referring to small thinly coated and large thinly coated from traffic particles. There is a contradiction regarding large thinly coated particles, related to solid fuel burning in line 238 but to traffic in line 254. Please clarify!

*Response*
*We wanted to make clear that we are not saying different modes of BC are exclusive to individual sources. Therefore, we have removed direct association of lightly coated mode as purely "traffic" and heavily coated particle as purely "solid fuel", from line 254 and rephrased line 238. As fluxes of heavily and lightly coated particle are discussed in section 4.5, that section also needed re-phrasing after this correction.*

*Correction*

*Line 258-261: Their study also suggested that thinly coated particles had strong association with traffic-related activities. Moderately coated and large thinly coated were related to combination of solid fuel (biomass and coal) and thickly coated particles were most associated with atmospheric aging.*

*Line 286-287: The implication of heavily coated particles in this study refers to the combination of thickly coated and moderately coated,  while lightly coated in this study refers to the combination of small thinly coated and large thinly coated particles defined and grouped by Liu et al. (2019).*

*Section 4.5: Rephrased.*

4) In the top panel of Figure 4a), there seems to be a systematic change in the difference between tower and ground-based measurements in the winter campaign from 25 November. Could you briefly discuss what might have happened? Also, in Figure 4b) please introduce the meaning of a and b as offset and slope or remove.

*Response*

*We add explanation for this systematic change between tower and ground, and also quantify the change. Also, updated Figure 4(b) as suggested by referee.*

*Correction*
*Line 266-269: Furthermore, during winter, we observe systematic difference between tower and ground measurement after data gap on 25th November, corresponding to instrument maintenance for ground SP2, which may have altered its performance. The difference was quantified by separating two periods and re-calculating slopes comparing the tower and ground measurements as shown in Figure 4(c). From this we can infer that the ground measurements increased by 24 %.*

*Updated Figure 4(b)*

5) Section 4.2 presents very interesting BC flux data, and I agree that while there is some uncertainty due to poor counting statistics, averaging can yield meaningful statistics. What is striking in Figure 6 is the much larger amplitude of net fluxes (both number and mass) in winter compared to summer. I recommend adding a brief passage that discusses the fraction of net deposition fluxes during the winter and summer campaigns, and the stronger net emission flux peaks in winter compared to summer.

*Response*
*We thank referee for their suggestion, and we have made following correction.*

*Correction*
*Line 325-329: The winter measurements are associated with higher uncertainty due to the lower atmospheric turbulence and the higher incidence of extreme values is consistent with this noise. As such, no individual 30-minute deposition events can be discerned with any statistical certainty. Therefore, we did not unpick such events/criteria and instead discussed averaged flux results in this study.*

6) In Figure 7, please add a 1 km scale and indicate the two major roads mentioned in the text to make the map more meaningful.

*Response*
*We have updated figure 7, major ring roads are in black and other roads in grey and 1 km scale is also added.*

*Correction*
*Figure 7 and caption updated.*

7) In Section 4.4, line 335, the authors state that "average flux values between the two seasons were similar, indicating a similar source between seasons". In my opinion, the second part is an overstatement.

*Response*
*We accept referee's suggestion and re-phrased the sentence.*

*Correction*
*Line 393-394: Average flux values between the seasons were similar, which is consistent with the source being the same in both seasons.*

8) Section 4.5 is very interesting and original, but I was left wondering whether or not seasonal differences occurred. I would strongly recommend to present Table 2 for the winter and summer seasons separately. This could also add to the discussion of a similar dominant source in the winter and summer seasons, which is also mentioned in the Conclusions in line 449.

*Response*
*As stated, the coating thickness data was not available for the summer dataset, so only data from the winter is presented. We have tried to make it clear that section 4.5 solely focuses on the winter measurements.*

*Correction*
*Here we update the table caption of Table 2.*

*"Winter season percentage values of concentrations and fluxes for coating classification and size classification. For coating classification, the total particles were separated according to coating contents using the threshold of Esca=3 for both BC mass and number. For size classification, the total particles were separated at Dc=180 nm, for BC mass and number. During summer, such analysis was not performed due to limitation of instrument at the time of experiment."*

9) In Table 2, use the term "heavily coated" instead of "highly coated" for consistency, and in the second column, exchange the associated sources given in parentheses, i.e. heavily coated (solid fuel) and lightly coated (traffic).

*Correction*
*We have applied this correction.*

**Technical comments:**

Put table captions above tables.
*"corrected"*

l. 39: Remove "." between "diseases" and "(".
*"Corrected"*

l. 39: "exposure limit" instead of "exposure limits".
*"Corrected"*

l. 43: Check units of "a 1 µm3 increase".
*"Corrected"*

l. 45: "with warming potential to the climate second only to CO2" - the term "warming potential" is confusing. I suggest using "radiative forcing" instead.
*"Corrected"*

l. 97: Remove "W." from reference Bond and Bergstrom (2006).
*"Corrected"*

l. 106: Remove "." between "Kondo" and "(".

*"Corrected"*

l. 164: "constant time-lag" instead of "constant time-lags".
*"Corrected"*

l. 169: Remove "." between "Sims" and "(".
*"Corrected"*

l. 212: "an instruments' response" instead of "an instruments response".
*"Corrected"*

l. 263: What do you mean by "precession limits"?
*"Corrected"*

l. 269: "with associated standard error" instead of "which associated standard error".
*"Corrected"*

l. 273: Remove ")" before ". The uncertainty..."
*"Corrected"*

l. 275: Remove "." between "Sims" and "(".
*"Corrected"*

l. 284: "before performing" instead of "before preforming".
*"Corrected"*

l. 314: "error bars represent" instead of "error bars represents".
*"Corrected"*

l. 315: "ensemble average" instead of "ensample average".
*"Corrected"*

l. 318: I recommend "for each hour of the day" instead of "for each time of the day".
*"Corrected"*

l. 330: Figure caption should read "Diurnal cycles for (a) number and (b) mass flux...".
*"Corrected"*

l. 346, Figure 9: What are the units of the color scale?
*"Corrected"*

l. 357: "heavily coated" instead of "highly coated".
*"Corrected"*

l. 360: Instead of "almost negligible", I recommend "smaller than 8 %". In my opinion, a contribution of 7.7 % is not negligible in this context.
"Corrected"

l. 379/380: "discussed by Squires et al. (2020)" instead of "discussed by (Squires et al., 2020)".
*"Corrected"*

l. 380: Values for BC/NOx ratios given in this line are not in agreement with values presented in Figure 10a).
*"Corrected"*

l. 388: Remove "the" after "in the literature for".
*"Corrected"*

Table 3: On p. 18, Table 3, I recommend adding the observed values for direct comparison.
*"Corrected"*

l. 398: In Table 3, in the third row, China V change "0.0057/0/009" to "0.0057/0.009", and in the last row, China III change "0.027/0/017" to "0.027/0.017"
*"Corrected"*

l. 408: Remove parentheses around "Zhang et al., 2015".
*"Corrected"*

l. 410: "BC emissions for the year 2013" instead of "BC emissions for year 2013".
*"Corrected"*

l. 436: "may reflect the limited suitability" instead of "may reflect the suitability".
*"Corrected"*

l. 459: Add "." between "inferred" and "The BC/NOx".
*"Corrected"*

l. 467: "This indicates" instead of "Indicating".
*"Corrected"*

l. 474: Remove "x" in "Atmxospheric". Interactive comment on Atmos. Chem. Phys.
*"Corrected"*

**Referee #2 Joshi et al. present a truly excellent example of how atmospheric chemical measurements can be used to directly improve models and provide new insight into underlying processes. The paper is well written, the data are robust, and the analysis is strong. This paper adds substantively to the literature. The flux uncertainty estimates, footprint analysis and tables of statistics are useful and well done. I recommend publication following minor revisions. Comments are below.**

paper Specific comments:

1) In the Methods, please be clear on the actual length of the inlet line: it's on a 102 m tower, but presumably longer? Figure 1 would be much improved with line lengths and, making the field setup more reproducible. Please provide estimates on particle losses from the inlet lines in the general size range. While those losses might not influence the calculated fluxes, size-dependent losses can cause bias in total particle mass or number fluxes – and some underestimates of emissions (or deposition).

*Response*
*The actual inlet line was about 115 m, that is added in the figure 1. Regarding line losses, we do not have measurement data to calculate line losses. However, as we had another SP2 measuring at ground level, BC concentration at two heights were examined in section 4.1. According to which, the upper limit of line losses would be 34 %, although the true losses are likely to be lower than this, as we expect concentrations at ground level to be higher.*

*Correction*
*Update on Figure 1.*

*Line 269-272: Furthermore, for tower measurements, there were also particle line losses, and the upper limit of line losses would be 34 %. This is based on the highest difference observed in winter seasons between two levels, although the true losses are likely to be lower than this, as we expect concentrations at ground level to be higher.*

2) A technical point, but Figure 2 would be easier to interpret if the x-axis ran from 0-24 hours (rather than ending at 18).

*Response/ correction*

*We have updated this figure, as suggested by referee.*

3) Figure 3 makes me worry about signal loss due to attenuation. While the authors write that "Given the majority of the flux is driven by much lower frequencies, a correction for high frequency loss is not applied." However, this fact is not clear from Figure 3. Please specify what percentage of the flux is driven by lower frequencies (and how that is defined), and what the influence of a correction would be. That is, would correcting the flux in Figure 3 for attenuation cause a 5% change (in which case, definitely minor), or a 30% change (in which case the corrections should be applied)?

*Response*

*As we are working with low S/N data it is tricky to quantify high frequency flux losses using covariance spectra in in Figure 3. Therefore, we have done further analysis using normalised Ogives, to quantify flux losses due to attenuation during both seasons. Using this approach flux losses in summer were quantified to be 9 % , however for winter measurement this approach did not fully work as the w'T Ogives were more damped compared w'BC' Ogives, which is not feasible and shows limitation of low S/N for this work. The correction was not applied as they small, and we could not quantify consistently in both seasons.*

*Correction*

*Line 233-240:* The high frequency flux loss due to attenuation was investigated using Ogive analysis (Spirig et al., 2005). In Figure 3(b), Ogives are calculated by cumulatively summing covariance values of wBC and wT (starting from lowest frequency) and normalised by total covariance. Similar to spectra analysis, in an ideal condition both wT and wBC Ogives would have same frequency distribution. In case of attenuation, difference could occur, which can be scaled at the low frequency, such that w'BC' Ogive collapses on w'T' Ogive. During summer, scaling of 0.91 is required, resulting flux loss of 9 %. In case of winter, as the wBC Ogive sits below w'T' and scaling of >1 required, suggesting w'T' spectrum is damped compared to w'BC' which is not physically feasible. Generally, this analysis describes challenges of emulating flux losses in noisy data, and as the losses are minor, we have not applied this correction.

[Figure]

[Figure]

4) The significant figures in Figure 4(b) and 10 are excessive. Please correct to something more reasonable. Also, it is not clear if 'a' is the slope or intercept in Figure 4b? Please specify on the figure what 'a' and 'b' refer to.

*Response/ Correction*
*We have resolved this issue and provided appropriate labels for Figure 4 (b) amd 10.*